# Preparation and Properties of Cyclodextrin Inclusion Complexes of *Hyperoside*

**DOI:** 10.3390/molecules27092761

**Published:** 2022-04-25

**Authors:** Xinyu Zhang, Jianqing Su, Xiaoya Wang, Xueyan Wang, Ruixue Liu, Xiang Fu, Ying Li, Jiaojiao Xue, Xiaoli Li, Rui Zhang, Xiuling Chu

**Affiliations:** College of Agronomy and Agricultural Sciences, Liaocheng University, Liaocheng 252000, China; xinxinxinxinyuyuyu@163.com (X.Z.); wangxy9625@163.com (X.W.); wangxueyan202203@163.com (X.W.); liuruixue919@163.com (R.L.); fuxiang2499570264@163.com (X.F.); ly15963373832@163.com (Y.L.); x15065441211@163.com (J.X.); lxl15006995983@163.com (X.L.); m17863709708@163.com (R.Z.)

**Keywords:** *Hyperoside*, inclusion complexes, antioxidant activity

## Abstract

In order to improve the aqueous solubility and enhance the bioavailability of *Hyperoside* (Hyp), three inclusion complexes (ICs) of Hyp with 2-hydroxypropyl-β-cyclodextrin (2H-β-CD), β-cyclodextrin (β-CD), and methyl-β-cyclodextrin (M-β-CD) were prepared using the ultrasonic method. The characterization of the inclusion complexes (ICs) was achieved using Fourier-transform infrared spectroscopy (FTIR), scanning electronic microscopy (SEM), X-ray powder diffraction (XRPD), thin-layer chromatography (TLC), and ^1^H nuclear magnetic resonance (^1^H NMR). The effects of the ICs on the solubility and antioxidant activity of Hyp were investigated. A Job’s plot revealed that the Hyp formed ICs with three kinds of cyclodextrin (CD), all at a 1:1 stoichiometric ratio. The FTIR, SEM, XRPD, TLC, and ^1^H NMR results confirmed the formation of inclusion complexes. The water solubility of the IC of Hyp with 2-hydroxypropyl-β-cyclodextrin was enhanced 9-fold compared to the solubility of the original Hyp. The antioxidant activity tests showed that the inclusion complexes had higher antioxidant activities compared to free Hyp in vitro and the H_2_O_2_–RAW264.7 cell model. Therefore, encapsulation with CDs can not only improve Hyp’s water solubility but can also enhance its biological activity, which provides useful information for the potential application of complexation with Hyp in a clinical context.

## 1. Introduction

*Hyperoside* (Hyp) is a natural flavonol glycoside complex, the chemical structure of which is shown in Figure 1. It is found in whole grasses of *Hypericaceae*, *Rosaceae*, *Platycodonaceae*, *Labiaceae*, and other plants [1]. Modern pharmacological studies have shown that Hyp has anti-inflammatory, antioxidant [2], anti-thrombotic, anti-fibrosis, analgesic, and other pharmacological effects [3,4]. It has therapeutic effects on cardiovascular and cerebral ischemia, liver fibrosis, heart failure, and other diseases [5,6]. The poor water solubility [7], low permeability, and instability of Hyp greatly affects its clinical application [8]. It is a useful tool to ameliorate the solubility of insoluble drugs, as well as to enhance their bioavailability through encapsulation technologies, such as in eutectic mixtures [9], microcapsule technology [10,11], and nanotechnology [12,13,14]. Among these approaches, cyclodextrin encapsulation has the advantages of low price, simple operation, mature technology, and good prospects [15,16,17].

CDs are oligosaccharide compounds with a circular, hollow, cylindrical structure resulting from amylose hydrolysis by cyclodextrin glucosyltransferase, usually containing six to eight D-(+)-glucopyranose units [18] which are connected by α-1, 4-glycosidic bonds. The common CDs are α-CD (containing six glucose residues), β-CD (containing seven glucose residues), and γ-CD (containing eight glucose residues) [19]. The molecular structure of cyclodextrin is that of a truncated conical oligosaccharide with hydrophobic cavities [20]. It can form water-soluble host–guest ICs or assemble complex supramolecular systems with many organic or inorganic molecules through various non-covalent interactions, such as van der Waals forces, hydrogen bonding, and hydrophobicity [21]. At present, cyclodextrin coating technology is widely used in food preservation [22,23], pharmaceutical preparations [24,25], environmental treatment [26,27], and other functional fields [28].

Studies have shown that flavonoids are compounds with a 2-phenylchromogenone structure, and the degree of matching between the size and shape of the cyclodextrin cavity and the difficulty of hydrogen bond formation plays a crucial role in their inclusion [29]. Therefore, it is better to select β-CD with a hydroxyl group or cyclodextrin with amino groups for encapsulation [30], as this is more conducive to the formation of hydrogen bonds between them. The most commonly used is β-CD, with a wide range of applications and low toxicity [31]. M-β-CD has a higher solubility than β-CD. Relevant literature shows that methylated cyclodextrin can improve the stability of ICs and has a good solubilization effect on insoluble drugs [32]. 2H-β-CD is the cyclodextrin with the most studied and best solubilization effect due to the oxygen atom of the hydroxyl group on the glucose residues being replaced with a hydroxypropyl group [33]. Studies have shown that solubility [34], stability [35], and bioavailability [36] of the drug are significantly improved after inclusion, and its pharmacological activity is also enhanced compared to that of the raw drug [37,38]. Danciu et al. [39] prepared inclusion compounds of rutin using β-CD and hydroxypropyl-β-cyclodextrin. The results showed that the ICs of rutin enhanced the antioxidant activity. Priyanka et al. [40] coated curcumin with β-CD. The water solubility and antioxidant activity of the ICs were significantly improved, and they found that the curcumin had the 5 h sustained release characteristic from the ICs. Hsu et al. [41] complexed the ethanol extract of rhubarb with 2H-β-CD, and the ICs showed both significantly increased water solubility and bioavailability, therefore, enhancing the anticancer effect in liver cells. To date, there is no literature on the ICs of Hyp and cyclodextrin, so it is of great significance to systematically evaluate these ICs in this study so as to expand their applications in medicine, food, and other fields in the future.

The preparation methods of ICs include the saturated solution, solution stirring, grinding, ultrasonic, and freeze-drying methods. The ultrasonic method applies the mechanical, cavitation, and thermal effects of ultrasonic, which can improve the molecular movement speed of the material, thereby increasing the penetration rate of medium molecules. In addition, the shock wave generated by ultrasonic wave propagation in liquid can release energy and promote the inclusion process of cyclodextrin and Hyp [42]. In this study, ICs of Hyp and cyclodextrin were prepared via the ultrasonic method, and the inclusion effect and properties of the ICs were evaluated through physicochemical characterization, including morphological observation, a water solubility test, TLC, ^1^H NMR, XPRD, thermal analysis, and an antioxidant test. This will lay the foundation for the clinical application of Hyp inclusion complexes.

## 2. Results

### 2.1. Determination of the Complexation Stoichiometry

The standard curve of the *Hyperoside* content determination is as follows:Y = 0.0373x + 0.0478, *R*^2^ = 0.9966.
where x (μg·mL^−1^) is the concentration of *Hyperoside*, and Y is the OD value.

The inclusion ratio between the three CDs and Hyp was determined via Job’s method [29]. The Job’s curves of 2H-β-CD, β-CD, and M-β-CD are shown in Figure 2. The highest point of the curve appeared when the mole fraction of the main body was 0.5, and it can be observed that the inclusion ratio of 2H-β-CD to Hyp was 1:1. Similar results were obtained with β-CD and M-β-CD, indicating that the inclusion ratio between the three types of cyclodextrin and Hyp was 1:1. Because in this experiment, the total concentration of the host and the guest was constant, and the concentration of Hyp varied with the concentration of cyclodextrin. In addition, cyclodextrin had no absorbance at 360 nm. Therefore, the greater the change of OD value of inclusion system, the stronger the inclusion efficiency of the host and the guest. This means that the inclusion effect of Hyp and 2H-β-CD was stronger than the other two types of CD.

### 2.2. Characterization

#### 2.2.1. SEM

According to the SEM images of the Hyp, CDs, three inclusion complexes, three physical mixtures, and powders (Figure 3), Hyp presented block crystals of different sizes (Figure 3a). 2H-β-CD showed a hollow, spherical structure, β-CD showed a small, block structure, M-β-CD was still seen to be spherical although its structure was incomplete, and all of the ICs were block crystals. However, the ICs of different types of cyclodextrin had different morphologies. In the Hyp–2H-β-CD inclusion complex, the spherical structure of 2H-β-CD itself completely disappeared, showing an irregular block and a smooth plane embedded in the shape of the small crystals, with obvious host–guest encircling (Figure 3b). The surface of the Hyp–β-CD inclusion complex was rough and had obvious cracks (Figure 3c). Only a small amount of guest embedding was observed on the surface of the Hyp–M-β-CD inclusion complex, and no cracks were observed on the surface (Figure 3d). The electron microscopic analysis of each IC showed that the volume of the IC was significantly larger than that of the monomer before inclusion and that the special morphological characteristic of Hyp did not exist. These obvious changes indicate that the ICs were effectively formed between Hyp and 2H-β-CD, β-CD, or M-β-CD, and the surface crystal structure changed after the inclusion complexes formed. In addition, each physical mixture still retained the structural characteristics of its host and guest, and a separate Hyp crystal and a separate cyclodextrin sphere could be observed. Thus, it could not be understood as a synthesis of a new substance [29].

#### 2.2.2. Thermogravimetric Analysis (TGA)/Differential Scanning Calorimetry (DSC)

The thermogravimetric curves of the Hyp, CDs, ICs, and their physical mixtures are shown in Figure 4a–c. Hyp–2H-β-CD had two procedures, the same as the dehydration of water molecules placed in the cavity and the decomposition of macrocycles. Hyp–2H-β-CD dehydration was under 50 °C with a 3.0% mass loss, and it lost 77.5% of its weight because of degradation under 317 °C [43]. The weight loss temperatures of Hyp–β-CD and Hyp–M-β-CD were 300 °C and 315 °C, respectively. According to the thermogravimetric curves of 2H-β-CD, β-CD, and M-β-CD, it can be seen that the TGA changes of the ICs and the TGA changes of physical mixtures were closely related to the existence of CDs. However, Hyp lost water at 100 °C and 170 °C and began to lose weight at 250 °C. Compared to Hyp, the ICs had higher thermal stability, as indicated by the greater temperature at which weight loss occurred.

The DSC thermograms of the Hyp, CDs, ICs, and their physical mixtures are shown in Figure 4d–f. The DSC curves of each physical mixture were similar to those of their respective CDs, which indicates that the inclusion of the Hyp and cyclodextrin could not take place by pour mixing them together. Hyp–2H-β-CD, Hyp–β-CD, and Hyp–M-β-CD showed evident peaks at temperatures of 238, 219, and 250 °C, respectively. These peaks could be due to the evaporation of the bound water in the polymers. The DSC results for Hyp uncovered two evident endothermic peaks at 123 and 190 °C. The endothermic peak at 123 °C can be attributed to a lack of water in Hyp; meanwhile, that at 190 °C can be attributed to the crystals melting, indicating the appearance of Hyp under a crystalline state [44]. However, the disappearing Hyp endothermal peaks in Hyp–2H-β-CD, Hyp–β-CD, and Hyp–M-β-CD indicate the encapsulation of Hyp in the CDs under an amorphous state. In contrast to materials with a crystal structure, the dissolving of materials with an amorphous structure does not need any energy to decompose the lattice; therefore, it is possible for them to diffuse easily via the polymeric matrix in order to achieve encapsulation of Hyp for continuous release.

#### 2.2.3. TLC Analysis

Thin-layer chromatography is a chromatographic method applied to segregate mixtures. Distinct complexes in sample mixtures move at distinct rates due to their different draws to the stationary phase, as well as their solubility in the solvent. Visualization of the TLC results of Hyp, 2H-β-CD, β-CD, M-β-CD, and the ICs, as well as their physical mixtures, under UV light is shown in Figure 5. Because there is no fluorescent with cyclodextrin, no spots were observed under UV light, while the three ICs and each physical mixture showed fluorescent spots for Hyp. Hyp and the physical mixture of Hyp/2H-β-CD moved at a faster speed than that of Hyp–2H-β-CD (Figure 5a), and a similar phenomenon also appeared in the inclusion system with β-CD as the host (Figure 5b). The difference of the movement velocity between Hyp and the ICs (Hyp–2H-β-CD and Hyp–β-CD) indicates that Hyp was encapsulated in a cyclodextrin cavity and formed larger molecules, which affected the moving velocity. The moving speed of the Hyp was faster than that of the physical mixtures, indicating that cyclodextrin in the physical mixture affected the Hyp movement. However, the moving speed of Hyp–M-β-CD was similar to that of Hyp with only a slight difference (Figure 5c), so it seems that its inclusion effect was relatively weak. The specific inclusion effect needs to be comprehensively evaluated from multiple aspects.

#### 2.2.4. XRPD Analysis

The changes of the characteristic peaks in the spectrum, such as an increase or decrease of the characteristic peaks, provide evidence of IC formation [45]. The XRPD test results of the monomer, ICs, and physical mixtures are shown in Figure 6. Hyp had a characteristic high-energy diffraction peak at 10°~40°, indicating that the drug had typical crystal structure characteristics. Some diffraction peaks of the Hyp and CDs were found in the physical mixtures, but the intensities of the peaks were weaker than that of the Hyp. However, the results of the ICs are obviously different from those of physical mixture in peak shape. The spectrum of the Hyp–2H-β-CD showed that the diffraction peak of Hyp disappeared, and only a wide diffraction peak at 15°~30° appeared. The changes in these peaks indicate that Hyp was included into the cyclodextrin [46,47]. The diffraction patterns of Hyp–β-CD and Hyp–M-β-CD still had the characteristic derivative peaks of the Hyp guest crystal type, but the intensity and number of peaks decreased significantly. All of the above changes were due to the fact that Hyp enters cyclodextrin when in the form of an ICs [48,49].

#### 2.2.5. FTIR Spectra

The FTIR spectra of Hyp, 2H-β-CD, the ICs, and their physical mixtures are illustrated in Figure 7. The FTIR spectrum of Hyp provided a large number of fingerprint absorption peaks, and the characteristic peaks of Hyp appeared (as shown in the gray-marked box). The 2H-β-CD FTIR spectrum showed peaks at 3394 cm^−1^ (O–H stretching vibration), 2927 cm^−1^ (C–H stretching vibration), 1639 cm^−1^, 1385 cm^−1^ (O–H plane bending), 1157 cm^−1^ (O–H stretching vibration), and 1034 cm^−1^ (C–O stretching/C–C stretching), as well as other strong absorption peaks. The infrared spectrum of Hyp–2H-β-CD showed similar structure with the infrared spectrum of 2H-β-CD. The peak position was basically consistent with that of 2H-β-CD. Compared to the free guest, the absorption intensity of the benzene ring and carbonyl chromophore in the infrared absorption spectrum of the IC was obviously weakened, which was due to the non-covalent weak force between the host and guest. The strong carbonyl absorption peak of Hyp at 1653 cm^−1^ appeared in the FTIR spectrum of the ICs, which is strong evidence that Hyp entered the 2H-β-CD cavity as a guest molecule, proving the formation of ICs between 2H-β-CD and Hyp. The other characteristic peaks of Hyp (wave numbers 1504 and 1259 cm^−1^, etc.) basically disappeared. In addition, Hyp–2H-β-CD showed strong absorption peaks of 3375, 2929, 1653, 1367, 1155, and 1032 cm^−1^, showing that the structure of 2H-β-CD was not damaged by the inclusion process. The peak of 2H-β-CD at 3375 cm^−1^ (O–H stretching vibration peak) was significantly stronger and wider than that at 3394 cm^−1^, indicating that an intermolecular hydrogen bond was formed between 2H-β-CD and Hyp. Moreover, the absorption peaks of 2H-β-CD (1034 and 1158 cm^−1^) moved to Hyp–2H-β-CD (1032 and 1155 cm^−1^), which was closely related to vibration (mainly C–C and C–O). The formation of a hydrogen bond led to a change of the bond electron density, which led to a change of the free frequency of stretching vibration, suggesting that a strong intermolecular hydrogen bond coaction occurred between the two. However, no new absorption band was generated in the spectrum of IC, and the formation process only took place as a physical interaction, without a change in the chemical structure. It can be seen from the above changes that the Hyp guest molecule entered the cavity of the host molecule 2H-β-CD instead of experiencing simple physical adsorption, indicating that the two formed a drug delivery system for the ICs. Some characteristic absorption peaks (wave numbers 3406, 2927, and 1655–1035 cm^−1^, etc.) of the host or guest were retained in the physical mixture of Hyp/2H-β-CD, which was obviously due to the simple addition of the spectra of the two, indicating that Hyp only existed in the mixture. It did not include into 2H-β-CD, which is consistent with the conclusions from literature [46].

#### 2.2.6. ^1^H NMR

In the inclusion process of host–guest molecules, the change in chemical shift is an effective method to characterize the ICs [50], and ^1^H NMR offers useful data on the spatial location of guest molecules in the cyclodextrin molecular cavity. After the IC was formed, the chemical shift of the H of the Hyp changed to varying degrees, as shown in Table 1 and Table 2. This means that the Hyp molecule successfully included into the 2H-β-CD cavity. The chemical shift of the H of the 2H-β-CD also changed, which may have been caused by the formation of hydrogen bond between the molecule of 2H-β-CD and Hyp. In addition, the chemical shift value of the H-5 in 2H-β-CD cavity was larger than that of H-3. In the stereoscopic structure of cyclodextrin, H-3 and H-5 are located in the cavity of cyclodextrin; H-3 is located at the large mouth end, and H-5 is located at the small mouth end. It is suggested that the Hyp enters the cavity of 2H-β-CD molecule from the small mouth end. The ^1^H NMR spectra of Hyp, 2H-β-CD, and Hyp–2H-β-CD are illustrated in Appendix A (see Appendix A). As shown in the figure, the characteristic peaks of Hyp were mainly concentrated between 6.20 and 7.68 ppm. The characteristic peaks of 2H-β-CD were mainly concentrated in the 3.49–4.98 ppm range, which illustrates that the ICs were successfully prepared.

### 2.3. Solubility Test

After the drug was included into cyclodextrin, the water solubility of the ICs was generally improved to varying degrees compared to that before inclusion, especially for substances that were insoluble in water themselves [16]. The test results are shown in Figure 8. The dissolution rate of Hyp in water was extremely slow, and the rate tended to become flat after 60 min of dissolution. Hyp–2H-β-CD dissolved rapidly in the first 10 min and then the dissolution rate decreased and became stable. After 10 min, the solubility of Hyp–2H-β-CD accounted for approximately 90.10% of the total solubility in the saturated state, which was higher than the other two kinds of IC. The solubilization of 2H-β-CD, β-CD, and M-β-CD increased the water solubility of Hyp to 1351.24, 1138.01, and 1040.42 µg·mL^−1^, respectively, which was remarkably higher than that in the Hyp monomer (approximately 153.09 µg·mL^−1^). In particular, Hyp–2H-β-CD had the strongest solubilization effect, up to 9-fold, which may be related to the intrinsic properties of 2H-β-CD. In order to realize its wide value in the field of clinical medicine, Hyp–2H-β-CD, which had the best solubilizing effect, was selected for follow-up tests to further explore its inclusion effect.

### 2.4. Effect of Temperature on the Complexation between Hyp and Cyclodextrin

The apparent stability constant (*K_s_*) of the host and guest were determined via UV–vis absorption spectrum titration. As shown in Figure 9a, when Hyp was in excess (10 mg), the absorption value of Hyp increased with the increasing concentration of 2H-β-CD. With the addition of 2H-β-CD, the absorption band of Hyp blue moved slightly at 355 nm, and the shoulder peak appeared at 360 nm. As the concentration of 2H-β-CD increased, the absorbance value of Hyp increased, and the color of the formed ICs deepened gradually with the increase in the Hyp concentration, as was visible to the naked eye (Figure 9c). These results confirmed that Hyp and 2H-β-CD formed an IC in the solution. In addition, the changes in the absorbance (∆Abs) of Hyp were researched based on increasing concentrations of 2H-β-CD under distinct temperatures. The results are shown in Figure 9b, indicating that the solubility of the ICs increased linearly as the temperature increased. The linearity of the plots indicates a 1:1 stoichiometry for the ICs.

Table 3 shows the intercept, slope, and *K_s_* of the straight-line graph of the solubility of Hyp/2H-β-CD at different temperatures. When the temperature was higher, the intercept of the linear curve was larger, indicating that the solubility of the Hyp aqueous solution increased with the increase in temperature. In addition, the *K_s_* decreased significantly with the increase in temperature, suggesting that the addition of Hyp to 2H-β-CD was accompanied by exothermic heat. Other complexes also showed similar properties. Mazzobre et al. [51] prepared an IC of α-terpinol and β-CD using the co-precipitation method. Mourtzinos et al. [52] incorporated thymol and geraniol with β-CD, and the solubility results showed that the above complex process was spontaneous and exothermic.

By studying the solubility at different temperatures, changes in enthalpy and entropy could be calculated. A Van ‘t Hoff diagram was formed by the Hyp and 2H-β-CD complex, as shown in Figure 10. ln*K_s_* had a linear relationship with the reciprocal of absolute temperature (1/*T*) (*R*^2^ = 0.997). The thermodynamic parameters could be counted from the slope and intercept of this linear curve: Δ*H* = −50.57 kJ mol^−1^ and Δ*S* = −117.02 J mol^−1^ k^−1^. Because Δ*H* was negative, it also indicates that the complexation of Hyp and 2H-β-CD was exothermic and of a low-energy type. Liu et al. [53] found that the occurrence of these complexations mainly depended on the spatial effect and hydrophobicity of the flavonoids. It may be that flavonoid molecules with stronger hydrophobicity replace water molecules in the cyclodextrin cavity and form hydrogen bonds with side groups in the cavity to further stabilize the inclusion structure. In addition, a negative value of Δ*S* indicated that the system became more ordered because the translational and rotational degrees of freedom of the complex molecule were lower than those of the free molecule [52]. According to Formula (3), the change in Gibbs free energy of the complexation interaction was calculated as follows: Δ*G*_30_ = −15.11 kJ mol^−1^, Δ*G*_40_ = −13.94 kJ mol^−1^, Δ*G*_50_ = −12.77 kJ mol^−1^. The negative value indicates that the complexation between Hyp and 2H-β-CD occurred spontaneously.

### 2.5. Determination of Antioxidant Activities

To date, a variety of detection methods have been developed based on chemical redox reactions to evaluate the antioxidant activities of flavonoids in traditional Chinese medicine. These detection methods can be roughly divided into two types according to the mechanism involved in achieving the antioxidant properties: single electron transfer or hydrogen atom transfer [54]. The reaction principles of the five detection methods used in this study are different. The 1,1-diphenyl-2-picrylhydrazyl (DPPH) method, based on hydrogen atom transfer, provides a hydrogen atom with the ability to quench free radicals by detecting antioxidants [55]. 2,2′amino-bis (3-ethyl-benzothiazoline sulfonate-6) ammonium salt (ABTS), reducing power, hydroxyl radical, and superoxide anion scavenging tests are typically performed by measuring the transfer of electrons by antioxidants to reduce the oxidability of free radicals or metal ions [56,57]. In addition, the scavenging ability of hydrophobic compounds (DPPH method) and the scavenging level of hydrophilic compounds (ABTS method) were detected in this study. Multiple methods were used to complement one another to fully evaluate the antioxidant ability of Hyp before and after coating. The DPPH test results (Figure 11a) showed that the DPPH radical scavenging activities increased with the increase in the Hyp, Hyp–2H-β-CD, and L-ascorbic acid concentrations in the range of 2.5–15 μg·mL^−1^ and showed dose dependency. When the sample concentration was added to 20 μg·mL^−1^, the scavenging rates were 91.14 ± 0.69%, 91.93 ± 0.40%, and 94.05 ± 0.22%, respectively. However, the scavenging activities of the DPPH free radical of the three samples tended to be flat with the increasing concentration. The IC_50_ of Hyp, Hyp–2H-β-CD, and L-ascorbic acid was 7.136, 6.267, and 8.194 μg·mL^−1^, respectively. In general, the DPPH radical scavenging ability of Hyp–2H-β-CD was stronger than that of uncoated Hyp, which may have been due to the fact that Hyp–2H-β-CD can release more hydrogen atoms bound to DPPH radicals in water [58].

In the ABTS radical scavenging assay (Figure 11b), the IC_50_ of Hyp, Hyp–2H-β-CD, and L-ascorbic acid was 6.427, 8.252, and 7.495 μg·mL^−1^, respectively. The radical scavenging ability of Hyp–2H-β-CD and Hyp on ABTS increased with the increase of sample concentration, and the scavenging ability of Hyp–2H-β-CD was always better than that of Hyp. When the concentration was 25 μg·mL^−1^, the maximum ABTS radical scavenging rate of Hyp and Hyp–2H-β-CD was 82.68 ± 0.62% and 91.61 ± 0.53%, respectively.

Considering that a hydroxyl radical, as the most active oxygen substance in organisms, easily reacts with other molecules to cause lesions, such as in lipids or proteins [59], it is essential to evaluate the scavenging ability of drugs on a hydroxyl radical. Since the solubility of Hyp in water can only reach 50 μg·mL^−1^, this study only compared the hydroxyl radical scavenging ability of Hyp and Hyp–2H-β-CD within this concentration range, and the results are illustrated in Figure 11c. Although the scavenging rates of Hyp–2H-β-CD and Hyp were only 24.67 ± 0.56% and 19.41 ± 0.29% at the highest sample concentration, respectively, the IC was still significantly better than the non-inclusion Hyp (*p* < 0.0001).

As can be seen from Figure 11d, the three samples all had certain scavenging effects on the superoxide anion, and the effects were positively correlated with the sample concentration. At the maximum concentration of 500 μg·mL^−1^, the scavenging ability of Hyp–2H-β-CD on superoxide anion was lower than that of L-ascorbic acid but higher than that of Hyp (*p* < 0.0001), with scavenging rates of 54.16 ± 0.24%, 98.47 ± 0.31%, and 41.46 ± 0.20%, respectively.

In addition, the amount of Fe^3+^ reduced to Fe^2+^ in potassium ferricyanide was detected to evaluate the reducing capacity, which is another potential indicator of the antioxidant capacity of a drug [60]. The results indicate that the absorbance value was higher and the reducing capacity was stronger. Figure 11e shows that the reducing ability of Hyp–2H-β-CD and Hyp was slightly lower than that of L-ascorbic acid. Although the IC showed stronger antioxidant activity than the non-inclusion Hyp, there was little influence on this index before and after inclusion.

It was concluded that the order of antioxidant capacity of the five antioxidant indexes from the test for oxidation resistance was L-ascorbic acid > Hyp–2H-β-CD > Hyp. Therefore, Hyp coated with 2H-β-CD showed better antioxidant activity than free Hyp, which laid a foundation for exploring its antioxidant activity at the cellular level. The antioxidant activity of the Hyp depends on the position and degree of hydroxylation in molecular structure and the free nature of ring structure. The formation of intramolecular hydrogen bonds and the presence of a second hydroxyl group enhance its antioxidant activity [61]. The enhanced antioxidant activity of the IC may be due, on the one hand, to the reduced free activity of the Hyp molecule in Hyp–2H-β-CD and the enhanced rigidity. On the other hand, the position of the hydroxyl group in the Hyp molecule is close enough to the hydroxyl group of 2H-β-CD to form intramolecular hydrogen bond in the IC, thus, enhancing its oxidation resistance.

### 2.6. Protective Effect of Hyp–2H-β-CD on H_2_O_2_–RAW264.7 Cells

The effects of Hyp and the IC of Hyp–2H-β-CD on the survival rate of RAW264.7 cells are shown in Figure 12a. Hyp and Hyp–2H-β-CD promoted the proliferation of RAW264.7 cells in the ranges of 50–125 and 50–175 μg·mL^−1^, respectively, and the proliferation-promoting effect first increased and then decreased with the increase in concentration. When the Hyp concentration reached 150 μg·mL^−1^, the survival rate of the RAW264.7 cells was remarkably lower than that of the blank group (*p* < 0.05), and, when the Hyp–2H-β-CD concentration reached 200 μg·mL^−1^, the survival rate of the RAW264.7 cells was also remarkably lower than that of blank group (*p* < 0.05). Therefore, low, medium, and high concentrations of Hyp–2H-β-CD at 50, 100, and 150 µg·mL^−1^ were selected for follow-up tests and were compared to 100 µg·mL^−1^ of Hyp with the strongest proliferative effect. The effect of H_2_O_2_ on the RAW264.7 cells is shown in Figure 12b. As the H_2_O_2_ concentration increased, the cell survival rate decreased correspondingly. When the concentration reached 1000 µM, the survival rate of the RAW264.7 cells was just 15.61 ± 3.10%, and the IC_50_ obtained was 493.21 µM. Therefore, the H_2_O_2_ concentration of 500 µM was determined as the experimental concentration. The protective effect of Hyp and Hyp–2H-β-CD on the H_2_O_2_-induced oxidative stress of RAW264.7 cells is shown in Figure 12c. The protective effect of 100 μg·mL^−1^ of Hyp–2H-β-CD was significantly greater than that of 100 μg·mL^−1^ of Hyp (*p* < 0.01). This means that Hyp coated with 2H-β-CD had a higher protective effect on H_2_O_2_–RAW264.7 cells. At the same time, the survival rate of 100 μg·mL^−1^ of Hyp–2H-β-CD group was higher than that of the blank group (*p* < 0.01).

Malondialdehyde (MDA), a lipid peroxidation metabolite, is formed by the degradation of polyunsaturated fatty acids by the active free radicals produced by oxygen metabolism and is an indicator of oxidative stress in cells [62]. Superoxide dismutase (SOD) can catalyze the decomposition of toxic superoxide anions (•O_2−_) into O_2_ and H_2_O_2_ and can prevent the toxic effects of superoxide anions from damaging cells [63]. SOD is an important antioxidant enzyme in the oxidative stress defense system. As shown in Figure 12d, the MDA content in the model group (1.14 ± 0.04 nmol·mg^−1^ prot) was remarkably higher than that in the blank group (0.46 ± 0.03 nmol·mg^−1^ prot) (*p* < 0.01), indicating that severe lipid peroxidation occurred in the cells after H_2_O_2_ stimulation. The level of MDA in the H_2_O_2_ + L-ascorbic acid group was remarkably lower than that in the model group (*p* < 0.01), but there was no evident distinction in the MDA content between Hyp and Hyp–2H-β-CD groups, indicating that Hyp and L-ascorbic acid had similar therapeutic effects. Compared to the Hyp group, the MDA content in the Hyp–2H-β-CD low-concentration group (*p* < 0.05) and the Hyp–2H-β-CD medium-concentration group (*p* < 0.01) was significantly decreased. In addition, the SOD activity in experimental groups was remarkably increased in comparison to the model group (*p* < 0.01), and the Hyp–2H-β-CD medium concentration group (24.89 ± 0.59 U·mg^−1^ prot) was higher than that in the Hyp group (22.25 ± 0.30 U·mg^−1^ prot) (*p* < 0.01). The SOD activity of the Hyp–2H-β-CD high concentration group was still remarkably distinct from that of the blank group (*p* < 0.01), as shown in Figure 12e. These results indicate that Hyp coated with 2H-β-CD could improve the content of antioxidant enzymes in RAW264.7 cells, which could inhibit the lipid peroxidation caused by H_2_O_2_ accumulation and reduce the oxidative stress of RAW264.7 cells. This result may be due to the special molecular structure, good stability, and easy chemical modification of CDs, showing great advantages in drug delivery. The host–guest complex structure formed by the 2H-β-CD and Hyp can protect the internal Hyp molecule to a certain extent, ensuring that it can better enter the cell to play its role after being dissolved in water.

## 3. Materials and Methods

### 3.1. Reagents and Instruments

*Hyperoside* (HPLC, ≥98%) was bought from Shanghai Yuanye Biotechnology Co., Ltd. (Shanghai, China). β-CD (AR, 98%), M-β-CD, 2H-β-CD (AR, ≥99.5%), absolute ethanol (AR, ≥99.7%), ethyl acetate (AR, ≥99.5%), glutaraldehyde (AR, 4%), methanol (AR, ≥99.9%) formic acid (AR, 85%), aluminum chloride (AR, 99%), potassium bromide (AR, 99%), and dimethyl sulfoxide (AR) were all bought from Shanghai Macklin Biochemical Co., Ltd. (Shanghai, China). Isoamyl acetate (AR, 99%) was bought from Shanghai Jizhi Biochemical Technology Co., Ltd. (Shanghai, China). GF254 silicone plate was bought from Qingdao Ocean Chemical Co., Ltd. (Shandong, China). L-ascorbic acid, DPPH, and ABTS were all purchased from Mall Beina Chuanglian Biotechnology Co., Ltd. (Henan, China). Distilled water was made by a lab water purification machine (model DBW-UP-10, Dongguan Dongbo Water Treatment Co., Ltd., Guangdong, China). RAW264.7 cells were bought from the Chinese Academy of Sciences Stem Cell Bank. Fetal bovine serum was purchased from Gibco, while 1640 medium was purchased from Hyclone Co., Ltd. (Logan City, UT, USA). A BCA protein concentration detection kit, a lipid oxidation (MDA) assay kit, and a total SOD activity assay kit were bought from Shanghai Biyuntian Biotechnology Co., Ltd. (Shanghai, China). An Acrodisc^®^ syringe filter with a Supor^®^ membrane was purchased from Pall (China) Co., Ltd. (Shanghai, China). The other reagents were all of at least analytical grade.

The instruments used were as followings: ultrasonic bath (Power-Sonic SB-600DTY; Ningbo Xinzhi Bio Technology Co., Ltd., Zhejiang, China), ultra-micro spectrophotometer (model F-1100, Hangzhou Suizhen Biotechnology Co., Ltd., Zhejiang, China), UV–visible spectrophotometer (model UV-5900, Shanghai Yuanxi Instrument Co., Ltd., Shanghai, China), carbon dioxide critical point dryer (model XD-1, Eiko Co., Ltd., Tokyo, Japan), ion gold plating instrument (model IB-3, Eiko Co., Ltd., Tokyo, Japan), thin-layer chromatography (model SB-2, Tianjin Tianfen Analytical Instrument Factory, Tianjin, China), centrifuges (model H1065-W, Hunan Xiangyi Laboratory Instrument Development Co., Ltd., Hunan, China), FTIR (model Thermo Nicolet iS5, Zequan International Group Shanghai Zequan Instrument Equipment Co., Ltd., Shanghai, China), constant-temperature oscillator (model HZQ-F160A, Shanghai Yiheng Scientific Instrument Co., Ltd., Shanghai, China), thermogravimetric analyzer (model Q500, TA Instruments Co., Ltd., New Castle, DE, USA), differential scanning calorimeter (model Q20, TA Instruments Co., Ltd., New Castle, DE, USA), X-ray diffractometer (model D8 Advance, Brook AXS, Karlsruhe, Germany), nuclear magnetic resonance NMR spectrometer (model AVANCE NEO 400M, Brook AXS, Karlsruhe, Germany), and scanning electron microscope (model JSM-840, JEOL Ltd., Tokyo, Japan).

### 3.2. Stoichiometry by Job’s Plot (Continuous Variation Method)

The *Hyperoside* content was measured according to the method described by Arya et al. with slight adjustments [40]. The concentrations of Hyp were 2.5, 5.0, 10.0, 20.0, and 40.0 μg·mL^−^^1^. The prepared samples were analyzed using the UV–vis spectrophotometric method at 360 nm using a spectrophotometer. The standard curve was established by plotting the absorbance value of the Hyp solution and its concentration.

The stoichiometric ratios of the ICs formed by Hyp with 2H-β-CD, β-CD, and M-β-CD were determined using Job’s method [64]. The total concentrations of Hyp and CD were kept constant (1 mmol·L^−1^), and a series of Hyp and CD mixtures were prepared so that the molar ratio of the two substances varied from 0.1 to 0.9. The absorbance value was measured at 360 nm after ultrasonic mixing at 25 °C for 60 min with ΔABS as the ordinate. ΔABS was the distinction in the absorbance of Hyp with CDs or not, and the ratio of [Hyp]/([Hyp] + [CD]) was taken as the abscissa.

### 3.3. Preparation of Physical Materials and ICs

Each physical mixture was mixed according to the molar ratio of the host and guest at 1:1. Hyp and three kinds of cyclodextrin were accurately weighed. They were thoroughly mixed in a ceramic mortar and stored at room temperature for later use.

According to the results of the Job’s curve, a complex of Hyp and CD was prepared at molar ratio of 1:1. First, 1.0 g of Hyp was dissolved with an appropriate amount of anhydrous ethanol, and another 30.82 g of 2H-β-CD was stirred to dissolve with 500 mL of water. The Hyp–ethanol solution was added drop by drop into 2H-β-CD solution under agitation, mixed for 1 h at an ultrasonic power of 300 W and an ultrasonic temperature of 50 °C and then shaken for 72 h. After cooling to room temperature, the undissolved solids were filtered using a 0.2 μm cellulose membrane. The filtrate was first frozen in a refrigerator at −80 °C and then transferred to a freeze-drying machine to obtain a yellow Hyp–2H-β-CD complex. The preparation of other inclusion complexes was similar to that of Hyp–2H-β-CD. The lyophilized samples were used for all feature analyses.

### 3.4. Properties of the ICs

#### 3.4.1. SEM

SEM is widely applied to measure the surface morphology and construction of materials. In SEM, an electron beam is used to scan the surface of the sample to obtain 3D spatial information [65]. This study compared the structural differences between *Hyperoside* powder, three kinds of IC, and their physical mixtures. The samples were fixed with 3% glutaraldehyde for 4 h, washed using a phosphoric acid buffer (0.1 M), and then dehydrated using an ethanol gradient of successive concentration (30%, 50%, 70%, 80%, 100%). Anhydrous ethanol was used twice, for 10 min each time. It was replaced by 50% and 100% isoamyl acetate on the first and second occasions, respectively. After drying with a critical point dryer, the powder uniformly adhered to a conductive adhesive and was mounted on an aluminum column before being sprayed with a gold coating. Imaging was performed using SEM at a 10 kV acceleration voltage and 600×, 2000×, or 5000× magnification.

#### 3.4.2. TGA and DSC

TGA using a Q500 was conducted in platinum pans with a nitrogen atmosphere (flow rate of 50 mL·min^−1^) at a heating rate of 5 °C·min^−1^ from 30 to 500 °C.

DSC analysis of the Hyp and inclusion complexes was carried out with a differential scanning calorimeter (Q20, TA Instruments) as depicted by Dai et al. [66] with modifications. The lyophilized sample (approximately 5.0 mg) was packed in a well-sealed Al pan and heated from 50 to 300 °C at a rate of 10 °C·min^−1^ under nitrogen gas.

#### 3.4.3. TLC

Sample solutions of 2H-β-CD, *Hyperoside*, and a Hyp–2H-β-CD complex were made by dissolving 33 mg of 2H-β-CD in 20 mL of water, 10 mg of Hyp in 20 mL of absolute ethanol, and 200 mg of the Hyp–2H-β-CD complex in 5 mL of water, separately. Subsequently, 10 mg of Hyp with 33 mg of 2H-β-CD dissolved in 20 mL of water was applied as a physical mixture solution of Hyp and 2H-β-CD. Four samples were applied to the same side of the GF254 silicone plate, and the bottom of the sample was immersed in the developing agent (ethyl acetate/formic acid/water = 30/3/2, *v*/*v*/*v*). When the upper end of the developing agent reached 1 cm from the upper end of the plate, the plate was removed. After drying, the image was developed by spraying with 5% aluminum trichloride ethanol solution and examined under a UV lamp (365 nm). The saturated solutions of the other two inclusion complexes were also tested as described above.

#### 3.4.4. XRPD

XRPD spectra of Hyp, Hyp–2H-β-CD, and the other ICs were obtain using a DX2700 diffractometer with Cu Kα radiation at a scanning rate of 5°·min^−1^. The powder sample was installed on a glass sample rack, and the scanning step was 2θ = 0.02°. The diffraction pattern was obtained at the angle of 5°~90° (θ–2θ).

#### 3.4.5. FTIR

FTIR spectrophotometry was carried out using a Thermo Nicolet iS5 instrument. First, 2 mg of the sample and 200 mg of potassium bromide were weighed and compressed into tablets. The blank control was made by pressing the potassium bromide powder into tablets. These were then placed into the liquid for detection. The *Hyperoside* monomer, 2H-β-CD, Hyp–2H-β-CD complex, and Hyp/2H-β-CD physical mixture were directly used for the measurements. Spectral information was obtained between 4000 and 400 cm^−1^ at room temperature.

#### 3.4.6. ^1^H NMR

The ^1^H NMR experiment was performed with a Bruker AVANCE NEO 400 M spectrometer at 298 K. First, 5 mg of the sample was added into the NMR tube. Next, 0.6 mL of methanol solvent was added into the tube containing Hyp. Following this, 0.6 mL of dimethyl sulfoxide solvent was placed into the tube with 2H-β-CD and Hyp–2H-β-CD. After the sample was dissolved to allow detection via a spectrometer, the ^1^H NMR spectrum was obtained.

### 3.5. Solubility Studies

An excess of Hyp (10 mg) or one of its ICs (250 mg) was added to 5 mL of water at 25 °C, and the amount of Hyp in the supernatant was determined every 5 min until the last measurement was performed at 200 min. Each sample was repeated three times to plot the dissolution curves with time of Hyp and its three ICs.

### 3.6. Effect of Temperature on Hyp–2H-β-CD

According to the method of Liu et al. [53] and Nguyen et al. [67], the influence of temperature on solubility was studied after appropriate adjustment. A series of molar concentrations of the 2H-β-CD solution (0,1, 2.5, 5, 10, 20, and 40 mM) was prepared in volumes of 5 mL, and excess Hyp (10 mg) was added to the centrifuge tubes of the above solutions. This was then encapsulated for 60 min at a power of 300 W and oscillated in a constant-temperature oscillator for 72 h. The equilibrium temperatures of the mixture were 30, 40, and 50 °C. After centrifugation, the supernatant was removed and filtered using 0.2 μm hydrophilic cellulose membrane filter to remove any undissolved Hyp. The UV–visible absorption spectra of the Hyp–2H-β-CD were tested using an ultra-micro spectrophotometer. By plotting the curve of Hyp solubility with the concentration of cyclodextrin, solubility maps of Hyp at different temperatures and concentrations of cyclodextrin were obtained. The *K_s_* was obtained from the solubility data, and the thermodynamics of coaction between Hyp and 2H-β-CD were analyzed. The relationship between the logarithm of the *K_s_* and the reciprocal of temperature was obtained. Experiments were all conducted thrice in parallel. The formula of *K_s_* was as follows:(1)$$K_{s}=\frac{\mathrm{slope}}{S\left(1-\mathrm{slope}\right)}$$
where *S* refers to the solubility of Hyp without cyclodextrin under different temperatures, and slope means the corresponding slope of the phase solubility diagrams.

On the basis of the Van‘t Hoff equation, namely Equation (2), the regression equation of ln*K_s_* on 1/*T* was used to obtain Δ*H* and Δ*S* and then the Gibbs free energy (Δ*G*) of the ICs was counted according to Equation (3):(2)$$\ln K_{s}=-\frac{\Delta H}{RT}+\frac{\Delta S}{R}$$
where *R* (J mol^−1^ K^−1^) is a universal gas constant, *T* (K) is the absolute temperature, Δ*H* (kJ mol^−1^) is the enthalpy change, and Δ*S* (J mol^−1^ K^−1^) is the entropy change.
(3)ΔG=ΔH−T×ΔS

### 3.7. Antioxidant Activity

DPPH, ABTS, the rate of hydroxyl radical scavenging, the rate of superoxide anion scavenging, and the reducing ability were used as indexes to compare the antioxidant capacity of Hyp before and after inclusion. A DPPH scavenging test was conducted according to the determination described by Andrade [68]. The ABTS radical scavenging test was as per the method of Aarland et al. [69]. The hydroxyl radical scavenging test was conducted according to the research method of Dai et al. [70], and the scavenging effect of the superoxide anion was determined via the pyrogallol autoxidation method [71]. The reduction capacity was determined by referring to the research method described by Chen [72]. Distilled water and L-ascorbic acid were used as blank and positive controls, respectively [73].

### 3.8. Antioxidant Test in H_2_O_2_–RAW264.7

#### 3.8.1. Drug Safety Evaluation

The drug safety concentration was evaluated using the CCK-8 method. The cell density in the logarithmic growth period was adjusted to 5 × 10^4^ mL^−1^, and the cells were cultured on 96-well plates at 200 µL per well. When the cells had grown to 80–90%, the supernatant was discarded and washed with PBS. Next, 200 µL of the medium containing Hyp and Hyp–2H-β-CD at distinct concentrations (75, 100, 125, 150, 175, and 200 µg·mL^−1^) was added, respectively, and the supernatant was discarded after the 24 h treatment. Next, 1640 medium containing 10% CCK-8 liquid was placed into each of the wells, which were then cultured at 37 °C for 30 min. The absorbance value was tested at 450 nm using a microplate reader, and the cell survival rate was counted based on the instructions.

#### 3.8.2. Establishment H_2_O_2_–RAW264.7 Model

The procedures described in Section 3.8.1 were repeated. After washing with PBS, we added 200 µL of serum-free 1640 medium containing different concentrations of H_2_O_2_ (100, 200, 400, 600, 800, and 1000 μM), followed by incubation for 2 h. The cell survival rate was determined by means of the CCK-8 method.

#### 3.8.3. Effect of Hyp and Its Inclusion Complex on the H_2_O_2_–RAW264.7 Cells

According to Section 3.8.1, the blank, H_2_O_2_, L-ascorbic acid (100 µg·mL^−1^), Hyp (100 µg·mL^−1^), and Hyp–2H-β-CD inclusion complex (50, 100, and150 µg·mL^−1^) groups were set. The blank and H_2_O_2_ groups used serum-free 1640 medium. After 24 h of drug treatment, the medium was removed and then, after washing, serum-free 1640 medium containing 400 µM of H_2_O_2_ was added for 2 h to calculate the cell survival rate.

#### 3.8.4. Detection of the MDA Activity and SOD Content

The cells were planted to six-well plate with 2 × 10^5^·mL^−1^, and, when the cells had grown and reached 80–90% of the well area, the supernatant was discarded and washed with PBS. The operation was carried out as in Section 3.8.3. The cells were lysed, and the supernatant was collected according to the kit instructions. The protein concentration was detected using the BCA method. The MDA activity and SOD content were detected strictly according to the instructions of the kit, and the relative content was calculated according to the protein concentration of the samples.

### 3.9. Data Analysis

Each group of experiments was carried out in triplicate, and the data are shown as the mean ± standard deviation. OriginPro 9.1 software was used for plotting, and SPSS 21 was used for multiple comparison and significance analysis.

## 4. Conclusions

*Hyperoside* exhibits a series of pharmacological activities, but its application is limited because of its poor water solubility and low bioavailability. In this paper, ICs of Hyp and cyclodextrin were successfully prepared with 1:1 stoichiometric. The evidence showed that the ICs with 2H-β-CD had higher water solubility compared with Hyp–β-CD or Hyp–M-β-CD. Thermodynamic studies demonstrated that the thermal stability of the three ICs was obviously stronger than that of Hyp, and the inclusion of Hyp with 2H-β-CD was a spontaneous exothermic process. According to SEM images and ^1^H NMR analysis, the Hyp molecules were inserted into the 2H-β-CD cavity to form the complex structure. The structure could protect Hyp from heat and prevent degradation at high temperature and significantly enhance its antioxidant capacity. The formation of this structure may be related to the presence of hydroxypropyl group in 2H-β-CD and significantly enhanced its capacity to entrap Hyp. This inclusion technology could be especially helpful for increasing the application of Hyp in functional foods, beverages, and drugs. It can be used for food preservation, especially for food with high fat content, and is expected to be widely used in food field instead of chemically synthesized antioxidants.

## Figures and Tables

**Figure 1 molecules-27-02761-f001:**
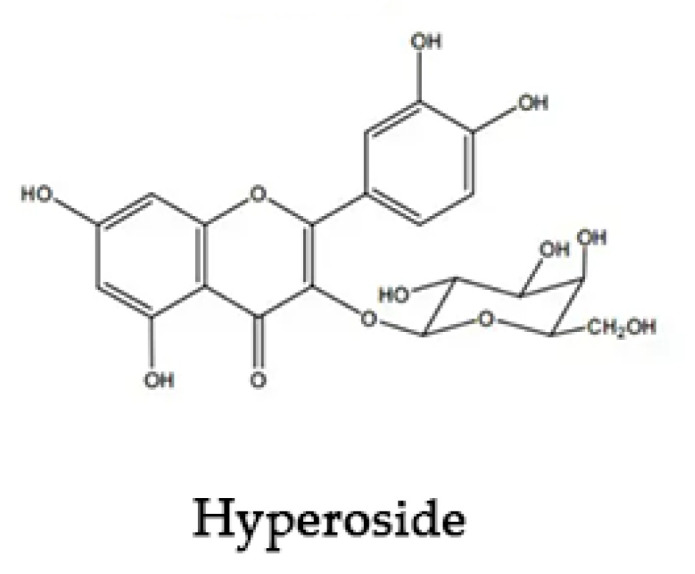
Chemical structure of *Hyperoside*.

**Figure 2 molecules-27-02761-f002:**
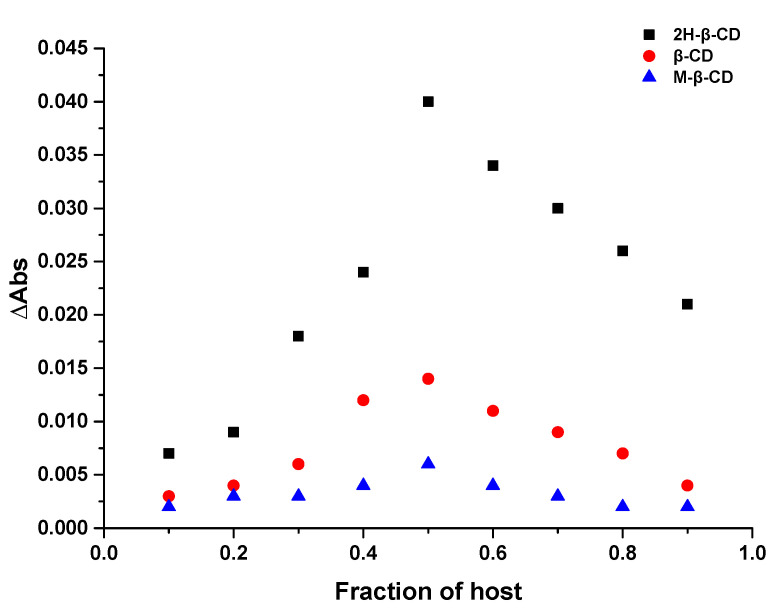
Job’s plot of 2H-β-CD, β-CD, and M-β-CD with Hyp.

**Figure 3 molecules-27-02761-f003:**
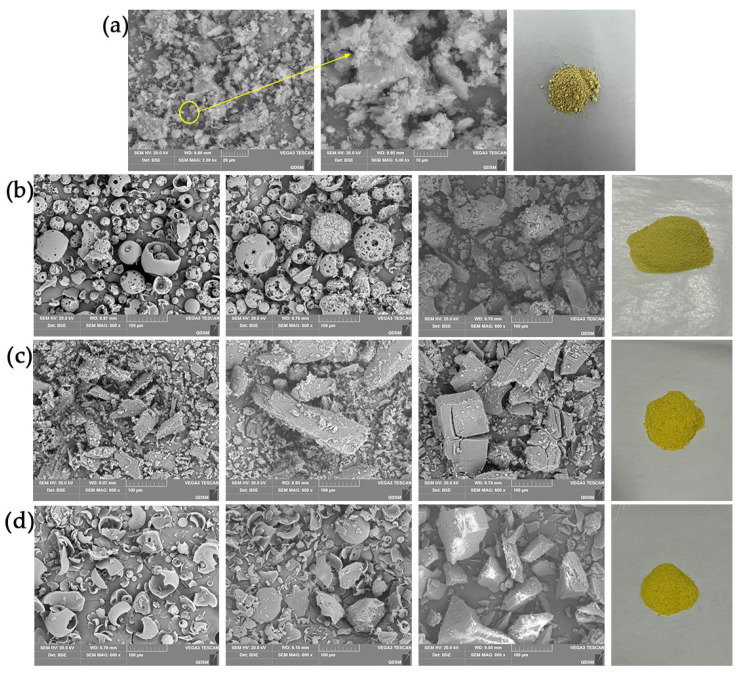
SEM image of Hyp with CD complex systems and their powder image. (**a**) Hyp (2000× and 5000× magnification) and powder image, (**b**) Hyp/2H-β-CD complex system, (**c**) Hyp/β-CD complex system, (**d**) Hyp/M-β-CD complex system. From left to right: Cyclodextrin image (600× magnification), physical mixture image (600× magnification), IC image (600× magnification), and powder image.

**Figure 4 molecules-27-02761-f004:**
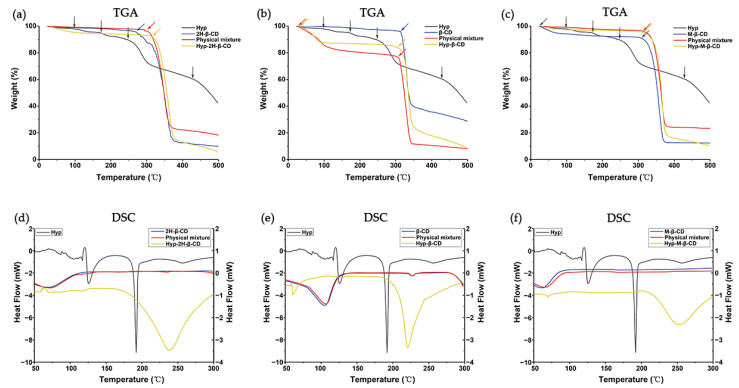
TGA and DSC analysis images of Hyp, CDs, ICs, and their physical mixtures. (**a**,**d**) Hyp/2H-β-CD complex system, (**b**,**e**) Hyp/β-CD complex system, (**c**,**f**) Hyp/M-β-CD complex system.

**Figure 5 molecules-27-02761-f005:**
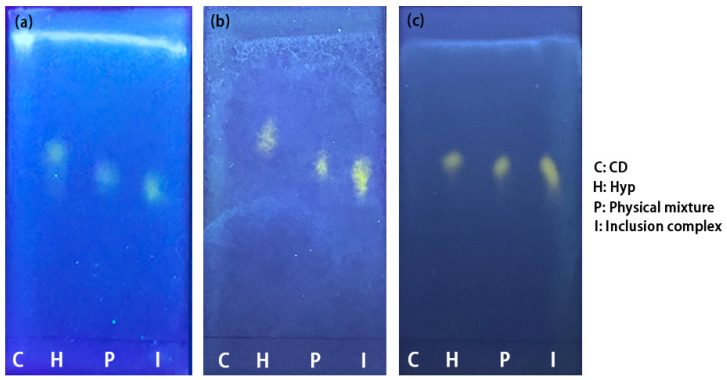
Thin-layer chromatography results of Hyp, CDs, ICs, and their physical mixtures under ultraviolet light. (**a**) Hyp/2H-β-CD complex system, (**b**) Hyp/β-CD complex system, (**c**) Hyp/M-β-CD complex system.

**Figure 6 molecules-27-02761-f006:**
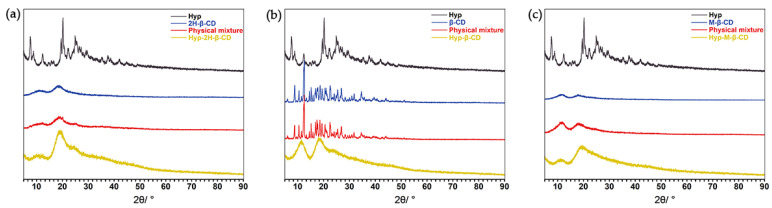
XRPD spectra of Hyp, CDs, physical mixtures, and the ICs. (**a**) Hyp/2H-β-CD complex system, (**b**) Hyp/β-CD complex system, (**c**) Hyp/M-β-CD complex system.

**Figure 7 molecules-27-02761-f007:**
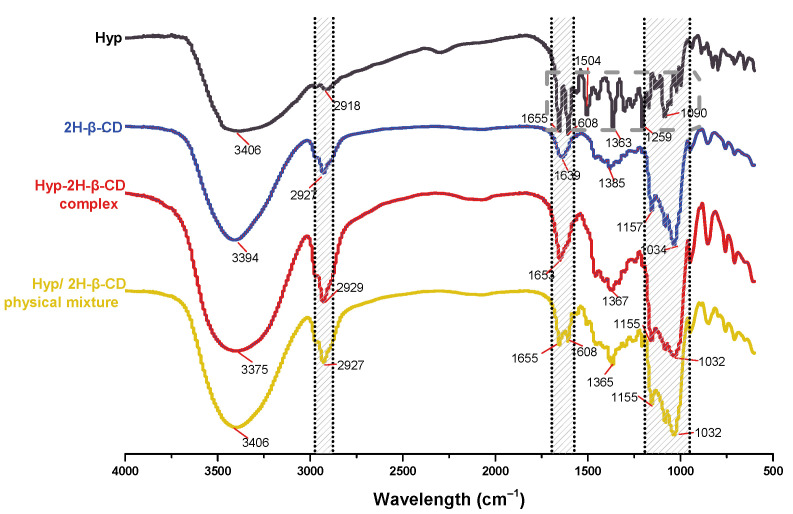
FTIR spectra of Hyp, 2H-β-CD, Hyp–2H-β-CD, and physical mixture.

**Figure 8 molecules-27-02761-f008:**
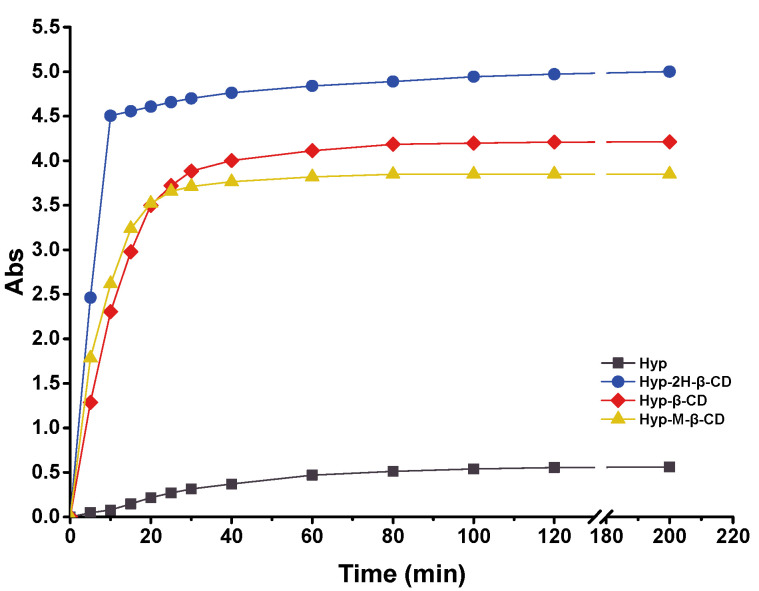
Solubility properties of Hyp, Hyp–2H-β-CD, Hyp–β-CD, and Hyp–M-β-CD in water.

**Figure 9 molecules-27-02761-f009:**
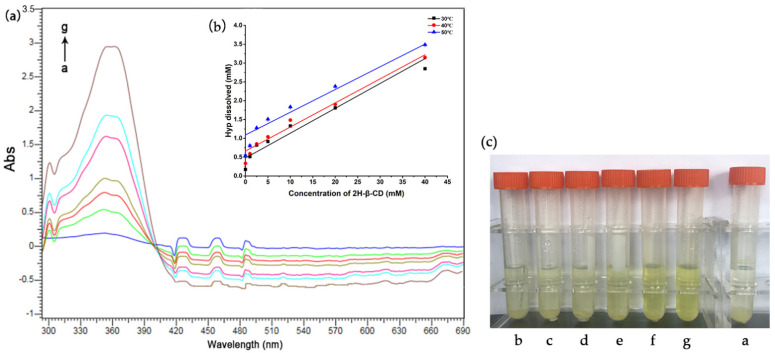
Diagram of the solubility of the Hyp and the IC. (**a**) UV–vis spectral changes of Hyp (sufficient) upon addition of 2H-β-CD at 40 °C; the concentration of 2H-β-CD (a–g): 0, 1, 2.5, 5, 10, 20, and 40 mM, respectively. (**b**) Solubility diagrams of Hyp with 2H-β-CD at 30, 40, and 50 °C. (**c**) Experimental diagram of the dissolution degree of Hyp in different concentrations of Hyp at 40 °C; the concentration of 2H-β-CD (a–g): 0, 1, 2.5, 5, 10, 20, and 40 mM, respectively.

**Figure 10 molecules-27-02761-f010:**
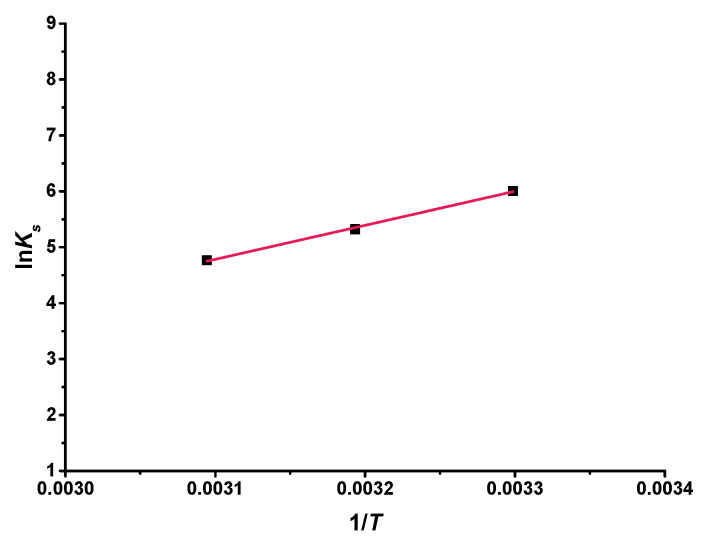
Van ‘t Hoff plot for the Hyp/2H-β-CD complex.

**Figure 11 molecules-27-02761-f011:**
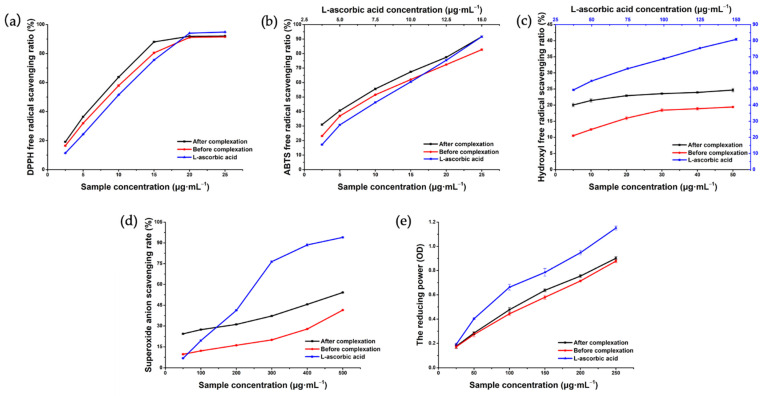
Antioxidant test of Hyp and Hyp–2H-β-CD. (**a**) DPPH radical scavenging activities; (**b**) ABTS radical scavenging activities; (**c**) hydroxyl radical scavenging activities; (**d**) scavenging activity for superoxide anion radicals; (**e**) the reducing power.

**Figure 12 molecules-27-02761-f012:**
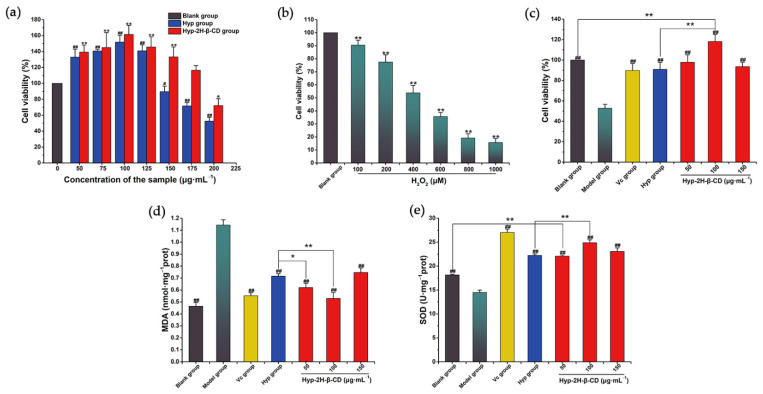
Results of the toxicity and antioxidant activity tests. (**a**) The effects of Hyp and Hyp–2H-β-CD on the cell viability of RAW264.7 cells. (**b**) The effects of H_2_O_2_ on RAW264.7 cells. (**c**) The protective effects of Hyp and Hyp–2H-β-CD on H_2_O_2_–RAW264.7 cells. (**d**) The effects on MDA content of Hyp and Hyp–2H-β-CD in the cells. (**e**) The effects on SOD content of Hyp and Hyp–2H-β-CD in the cells. In (**c**–**e**), model group (H_2_O_2_, 500 µM), L-ascorbic acid group (L-ascorbic acid, 100 µg·mL^−1^), Hyp group (Hyp, 100 µg·mL^−1^), and Hyp–2H-β-CD group (50, 100, 150 µg·mL^−1^). Note: In (**a**,**b**), compared to the blank group, * indicates evident distinction at *p* < 0.05, ** shows very evident distinction at *p* < 0.01, ^#^ indicates significant difference at *p* < 0.05, and ^##^ shows very evident distinction at *p* < 0.01. In (**c**–**e**), for the comparison between the two groups, * indicates evident distinction at *p* < 0.05, ** indicates extremely evident distinction at *p* < 0.01; compared to the model group, ^##^ indicates extremely evident distinction at *p* < 0.01.

**Table 1 molecules-27-02761-t001:** ^1^H NMR chemical shifts of Hyp before and after interaction with 2H-β-CD.

*Hyperoside* (H)	*Hyperoside* (δ_0_)	Hyp–2H-β-CD (δ_1_)	Δδ_1_
H-2′	7.68	7.71	0.03
H-6′	7.52	7.45	0.07
H-5′	6.82	6.99	0.17
H-6	6.20	6.21	0.01
H-8	6.40	6.67	0.27

**Table 2 molecules-27-02761-t002:** ^1^H NMR chemical shifts of 2H-β-CD before and after forming IC.

2H-β-CD (H)	2H-β-CD (δ_0_)	Hyp–2H-β-CD (δ_1_)	Δδ_1_
H-1	4.97	4.98	0.01
H-2	3.52	3.54	0.02
H-3	3.78	3.77	0.01
H-4	3.50	3.49	0.01
H-5	3.75	3.77	0.02
H-6	3.55	3.56	0.01

**Table 3 molecules-27-02761-t003:** The intercept, slope, and *K_s_* at different temperatures were obtained from the Hyp/2H-β-CD solubility diagram.

Temperature (°C)	Intercept (mM)	Slope	*K_s_* (M^−1^)
30	0.477	0.066	406.3
40	0.662	0.064	204.8
50	1.091	0.060	117.4

## Data Availability

Not applicable.

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
