# Peer review of "Preparation and Properties of Cyclodextrin Inclusion Complexes of Hyperoside"

_molecules, 2022, doi:10.3390/molecules27092761_

Round 1
Reviewer 1 Report
The paper “Preparation and properties of cyclodextrin inclusion complexes of Hyperoside” by Zhang et al. deals with the characterization of inclusion complexes of hyperoside in beta-cyclodextrin, and two of its derivatives. The authors use a number of experimental techniques for their study, and conclude that the complex with the hydroxypropyl variant of beta-cyclodextrin is particularly effective at improving solubility and enhancing biological activity.
Overall, the study is interesting and the techniques employed by the authors provide a very comprehensive characterization of the inclusion complexes. However, the quality of English should be improved in the final version of the manuscript, as the work is hard to read at the current stage. The following points should also be taken into account before publication:
Materials and methods:
Eq. 1: please define exactly what you mean by ‘slope’
line 245, page 5: please define the abbreviation Vc
Please define the abbreviations MDA and SOD the first time you mention them
Results:
line 284, page 6: Y=0.0373x+0.0478: is x the concentration in ug/ml, and Y the OD value? Please clarify.
Line 287, page 6: “Figure. 1a” should be Figure 1
Line 381, page 10: “[…] presenting amorphous or amorphous form” please revise this sentence. “Zheng” should be “Zheng et al."
Please explain why the FT-IR and NMR analyses were performed only for the H-b-CD complex, and not for the b-CD/M-b-CD ones.
Figure 7: figures are a bit blurry. Would it be possible to improve resolution?
Line 445, page 12: “[…] that were insoluble or insoluble in water themselves” please rephrase
Line 493, page 14: “Ks had a linear relationship […]” maybe the authors meant that lnKs had a linear relationship?
Line 502, page 14: “In addition, a negative value of ΔS indicated that the system became more stable […]” I understand what the authors want to say here, i.e., that the system became more ordered, but please be careful about the proper use of words. A system with entropy decrease cannot be more stable, because systems at higher entropy are actually more stable thermodynamically.
Lines 535-536, page 15: “the ABTS scavenging ability of Hyp-2H-β-CD as well as Hyp was always better than that of Hyp” please rephrase
Lines 537-538, page 15: “[...]the maximum ABTS radical scavenging rate of Hyp as well as Hyp-2H-β-CD was 91.61±0.53% and 82.68±0.62%, respectively” I think the numbers for Hyp and Hyp-2H-β-CD are inverted. Please verify.
Line 549, page 16: “Hyp-2H-β-CD had a higher scavenging capacity than that of L-ascorbic acid” this seems not to be true from Figure 11d. Please verify.
Figure 12: please better define what you mean by model group and Vc group.
Author Response
Dear reviewer,
Thank you very much to review my manuscript (molecules-1669537), and give me some suggestions. These comments are valuable and very helpful. We have read through comments carefully and have made corrections. Based on the instructions you provided, we uploaded the field of the revised manuscript. Revisions in the text are shown using red highlight for additions, and strike through font for deletions. The responses to your comments are marked in red and presented following in this word.
We would love to thank you for allowing us to resubmit a revised copy of the manuscript and we highly appreciate your time and consideration.
Yours sincerely,
Xinyu Zhang
General Comments:
Comment 1: please define exactly what you mean by ‘slope’
Response 1: A supplementary description “and slope means the corresponding slope of the phase solubility diagrams” has been added to the corresponding position in the article. Line 693
Comment 2: line 245, page 5: please define the abbreviation Vc
Response 2: The whole text uniformly replaces the word "Vc" with the word "L-ascorbic acid".
Comment 3: Please define the abbreviations MDA and SOD the first time you mention them
Response 3: A supplementary description “Malondialdehyde (MDA)” and “Superoxide Dismutase (SOD)” have been added to the corresponding position in the article. Line 500
Comment 4: line 284, page 6: Y=0.0373x+0.0478: is x the concentration in ug/ml, and Y the OD value? Please clarify.
Response 4: A supplementary description “where, x (μg·mL-1) is the concentration of Hyperoside and Y is OD value.” have been added to the corresponding position in the article. Line 118
Comment 5: Line 287, page 6: “Figure. 1a” should be Figure 1
Response 5: Change “Figure. 1a” to “Figure 2”. Line 121
Comment 6: Line 381, page 10: “[…] presenting amorphous or amorphous form” please revise this sentence. “Zheng” should be “Zheng et al.”
Response 6: The sentence was removed. Line 246
Comment 7: Please explain why the FT-IR and NMR analyses were performed only for the H-b-CD complex, and not for the b-CD/M-b-CD ones.
Response 7: We initially performed scanning electron microscopy, thermogravimetry, thermal analysis, thin layer chromatography and X-ray diffraction analysis on these three inclusion complexes. The results showed that the three cyclodextrins formed inclusion complexes with Hyperoside. Subsequently, we carried out Solubility test, among which the solubilization effect of Hyp-2H-β-CD was selected as the research object of the follow-up test. Therefore, we only performed complementary experiments on it in the FTIR and NMR analyses.
Comment 8: Figure 7: figures are a bit blurry. Would it be possible to improve resolution?
Response 8: In order to present the test results more clearly, we presented the test results in the form of tables, as shown in Table 1 and 2. The spectrogram was moved into the supplementary material, as shown in Figure S1.
A supplementary description “After the IC was formed, the chemical shift of H of the Hyp changed to varying degrees, as shown in Table 1 and 2. This means that the Hyp molecule successfully included into the 2H-β-CD cavity. The chemical shift of H of the 2H-β-CD also changed, which may be caused by the formation of hydrogen bond between the molecule of 2H-β-CD and Hyp. In addition, the chemical shift value of H-5 in 2H-β-CD cavity is larger than that of H-3. In the stereoscopic structure of cyclodextrin, H-3 and H-5 are located in the cavity of cyclodextrin, and H-3 is located at the large mouth end and H-5 is located at the small mouth end. It is suggested that the Hyp enters the cavity of 2H-β-CD molecule from the small mouth end.” have been added to the corresponding position in the article. Line 298
Comment 9: Line 445, page 12: “[…] that were insoluble or insoluble in water themselves” please rephrase
Response 9: Change “that were insoluble or insoluble in water themselves” to “that were insoluble in water themselves”. Line 329
Comment 10: Line 493, page 14: “Ks had a linear relationship […]” maybe the authors meant that lnKs had a linear relationship?
Response 10: Change “Ks had a linear relationship” to “lnKs had a linear relationship”. Line 383
Comment 11: Line 502, page 14: “In addition, a negative value of ΔS indicated that the system became more stable […]” I understand what the authors want to say here, i.e., that the system became more ordered, but please be careful about the proper use of words. A system with entropy decrease cannot be more stable, because systems at higher entropy are actually more stable thermodynamically
Response 11: Many thanks for your reminding, many similar non-standard words in the article have been rewritten. I will also strive to improve my writing skills in the future. Change “In addition, a negative value of ΔS indicated that the system became more stable” to “In addition, a negative value of ΔS indicated that the system became more ordered”. Line 391
Comment 12: Lines 535-536, page 15: “the ABTS scavenging ability of Hyp-2H-β-CD as well as Hyp was always better than that of Hyp” please rephrase
Response 12: Sentence “With the increase of sample concentration, the ABTS scavenging ability of Hyp-2H-β-CD as well as Hyp was always better than that of Hyp.” is changed to Sentence “The radical scavenging ability of Hyp-2H-β-CD and Hyp on ABTS increased with the increase of sample concentration, and the scavenging ability of Hyp-2H-β-CD was always better than that of Hyp.”. Line 430
Comment 13: Lines 537-538, page 15: “[...]the maximum ABTS radical scavenging rate of Hyp as well as Hyp-2H-β-CD was 91.61±0.53% and 82.68±0.62%, respectively” I think the numbers for Hyp and Hyp-2H-β-CD are inverted. Please verify.
Response 13: Sentence “the maximum ABTS radical scavenging rate of Hyp as well as Hyp-2H-β-CD was 91.61±0.53% and 82.68±0.62%, respectively.” is changed to Sentence “the maximum ABTS radical scavenging rate of Hyp and Hyp-2H-β-CD was 82.68 ± 0.62% and 91.61 ± 0.53%, respectively.”. Line 434
Comment 14: Line 549, page 16: “Hyp-2H-β-CD had a higher scavenging capacity than that of L-ascorbic acid” this seems not to be true from Figure 11d. Please verify.
Response 14: Sentence “Hyp-2H-β-CD had a higher scavenging capacity than that of L-ascorbic acid (p < 0.0001)” is changed to Sentence “the scavenging ability of Hyp-2H-β-CD on superoxide anion was lower than that of L-ascorbic acid but higher than that of Hyp (p < 0.0001)”. Line 448
Comment 15: Figure 12: please better define what you mean by model group and Vc group
Response 15: A supplementary description " In c, d, e, Model group (H2O2, 500 µM), L-ascorbic acid group (L-ascorbic acid, 100 µg·mL-1), Hyp group (Hyp, 100 µg·mL-1) and Hyp-2H-β-CD group (50、100、150 µg·mL-1)." has been added to the corresponding position in the article. Line 535

Reviewer 2 Report
Manuscript ID: molecules-1669537
Title: Preparation and properties of cyclodextrin inclusion complexes of
Hyperoside
Hyp and βCd derivatives are used to characterize physicochemical properties. For the inclusion of compounds, their antioxidant capacity is also being evaluated. Since Hyp is a food ingredient, it is expected to be a cornerstone of future research and development.
Some corrections are expected.
1) Line 29
Can you suggest a structure for Hyperoside (Hyp)? This is to make it possible for other researchers to visualize what the structure is like.
2) Line 139
Regarding Job's plot, the sample preparation before measuring absorbance is treated with ultrasonic treatment for 60 min. What is the reason for 60 min?
3) Line 188
What are the operating conditions? For example, using Cuα?
Please add.
4) Line281.
With respect to Job's plot (Fig. 1), Δabs is higher with 2HP-βCD than with other M-βCD and βCⅮ in this study, according to Fig. 1. This means that the dissolution of Hyp is also better for CDs. Why is that?
5) Line322
In the TG curves, why don't you mention the onset temperature of weight loss of Hyp-CDs in the text?
6) In SEM, TG, XRD and IR, βCD, M-βCD and 2HP-β No data are shown for CD.
Therefore, if it is SEM, we only compare Hyp to the prepared Hyp-βCDs (3 types) and do not consider comparing Hyp to CDs. The same is true for TG, DSC, IR, and XRD. In the first place, it should be shown as CDs. Since the purpose of this study is characterization, please provide the data to be compared.
7) Line559
You are of the opinion that "oxidation resistance was L-ascorbic acid >Hyp-2H-β-CD >Hyp. Why is the difference in the Hyp antioxidant capacity when it is included?
Hyp is present in the CD cavity when it is included. Please discuss antioxidant capacity and inclusion mode.
8) Finally, in Your opinion, how about suggesting a little more detail on how this research will affect the expansion of its application in the food field?
Fruitful collaboration shall continue. Thanking you!
Author Response
Dear reviewer,
Thank you very much to review my manuscript (molecules-1669537), and give me some suggestions. These comments are valuable and very helpful. We have read through comments carefully and have made corrections. Based on the instructions you provided, we uploaded the field of the revised manuscript. Revisions in the text are shown using red highlight for additions, and strike through font for deletions. The responses to your comments are marked in red and presented following in this word.
We would love to thank you for allowing us to resubmit a revised copy of the manuscript and we highly appreciate your time and consideration.
Yours sincerely,
Xinyu Zhang
General Comments:
Comment 1: Line 29 Can you suggest a structure for Hyperoside (Hyp)? This is to make it possible for other researchers to visualize what the structure is like.
Response 1: I think your suggestion is very necessary, which will be conducive to a clearer understanding of the research object. Therefore, I added the chemical structure diagram of Hyperoside in the introduction part, as shown in Figure 1.
Comment 2: Line 139
Regarding Job's plot, the sample preparation before measuring absorbance is treated with ultrasonic treatment for 60 min. What is the reason for 60 min?
Response 2: This study is slightly different from that of other cyclodextrin inclusion complexes. In this study, the inclusion complex was prepared by ultrasonic method. In the preliminary experiment, we concluded that the inclusion effect of 60 min was better under ultrasonic conditions, but the inclusion effect could not be significantly improved even if the time was prolonged. Therefore, this test condition is established as inclusion condition. Because this study focused on the characterization of each inclusion complex and the comparison of the efficacy of Hyperoside before and after inclusion, the inclusion conditions were not explained in detail. I sincerely hope you can understand my original intention.
Comment 3: Line 188 What are the operating conditions? For example, using Cuα? Please add
Response 3: A supplementary description " XRD spectra of Hyp, Hyp-2H-β-CD, and the other ICs were obtain using a DX2700 diffractometer with Cu-Kα radiation at a scanning rate of 5°·min-1." has been added to the corresponding position in the article. Line 648
Comment 4: Line281. With respect to Job's plot (Fig. 1), Δabs is higher with 2HP-βCD than with other M-βCD and βCⅮ in this study, according to Fig. 1. This means that the dissolution of Hyp is also better for CDs. Why is that?
Response 4: Job's figure did show that the inclusion effect of Hyperoside and 2H-β-CD was stronger than the other two kinds of cyclodextrins. Because in this experiment, the total concentration of the host and the guest is determined, and the series of concentration gradients of Hyperoside is also determined and constant. In addition, each cyclodextrin has no absorbance at 360nm. Therefore, the greater the CHANGE of OD value of inclusion system, the stronger the inclusion efficiency of the host and the guest. However, due to the small difference in ΔAbs of the three inclusion systems at the highest point, the inclusion effect of Hyperoside and 2H-β-CD was not emphasized here in the manuscript, but reflected by the subsequent solubilization effect. This conclusion has now been added to the manuscript.
A supplementary description " Because in this experiment, the total concentration of the host and the guest is constant, and the concentration of hyp varies with the concentration of cyclodextrin. In addition, cyclodextrin has no absorbance at 360 nm. Therefore, the greater the change of OD value of inclusion system, the stronger the inclusion efficiency of the host and the guest. This means that the inclusion effect of Hyp and 2H-β-CD is stronger than the other two types of CDs." has been added to the corresponding position in the article. Line 126
Comment 5: Line322 In the TG curves, why don't you mention the onset temperature of weight loss of Hyp-CDs in the text?
Response 5: A supplementary description " The weight loss temperatures of Hyp-β-CD and Hyp-M-β-CD were 300 °C and 315 °C, respectively." has been added to the corresponding position in the article. Line 170
Comment 6: In SEM, TG, XRD and IR, βCD, M-βCD and 2HP-β No data are shown for CD.
Therefore, if it is SEM, we only compare Hyp to the prepared Hyp-βCDs (3 types) and do not consider comparing Hyp to CDs. The same is true for TG, DSC, IR, and XRD. In the first place, it should be shown as CDs. Since the purpose of this study is characterization, please provide the data to be compared.
Response 6: The three groups of SEM micrographs (b) (c) (d) images from left to right are cyclodextrin diagram (600× magnification), physical mixture diagram (600× magnification), complex diagram (600× magnification) and powder diagram respectively. The results of TGA, DSC, and XRD of cyclodextrins and three physical mixtures are added to the text. Section 2.2.2 and 2.2.4; Figure 4 and 6
Comment 7: Line559
You are of the opinion that "oxidation resistance was L-ascorbic acid >Hyp-2H-β-CD >Hyp. Why is the difference in the Hyp antioxidant capacity when it is included?
Hyp is present in the CD cavity when it is included. Please discuss antioxidant capacity and inclusion mode.
Response 7: A supplementary description " The antioxidant activity of the Hyp depends on the position and degree of hydroxylation in molecular structure and the free nature of ring structure on the other hand. The formation of intramolecular hydrogen bonds and the presence of a second hydroxyl group enhance its antioxidant activity [62]. The enhanced antioxidant activity of the IC may be due on the one hand to the reduced free activity of the Hyp molecule in Hyp-2H-β-CD and the enhanced rigidity. On the other hand, the position of the hydroxyl group in the Hyp molecule is close enough to the hydroxyl group of 2H-β-CD to form intramolecular hydrogen bond in the IC, thus enhancing its oxidation resistance." has been added to the corresponding position in the article. Line 463
Comment 8: Finally, in Your opinion, how about suggesting a little more detail on how this research will affect the expansion of its application in the food field?
Response 8: First of all, I think the issue of food preservation is more important in the food field. Many food additives with antibacterial properties, such as cinnamaldehyde, have the disadvantage of having a pungent odor and strong volatility, and are less antibacterial when present in emulsion form. Recently, it has been reported that the inclusion of cinnamaldehyde with cyclodextrin can reduce its irritating odor and volatilization loss, and can transform the liquid into a solid powder for easy use. For example, the coating treatment of fresh-cut melon with trans-cinnamaldehyde cyclodextrin inclusion complex mixed with chitosan can effectively maintain the hardness and color of melon, reduce weight loss and extend shelf life. I believe that if it is used for the preservation of meat, it may also achieve good storage quality. Secondly, biscuits, bread, cakes and other baked goods generally need to go through 170~300℃ high temperature baking, in this process, the essence is easy to be destroyed or volatilized. I learned that if the essence is microencapsulated with cyclodextrin as the carrier, it can protect the essence. The inclusion complex has a certain tolerance to high temperature conditions, and only under high temperature conditions can the cell wall rupture to release the essence. Therefore, the inclusion of cyclodextrin can also be used to prepare high temperature resistant flavor suitable for baking food processing. Because Hyperoside also has strong antioxidant properties, its inclusion complex can also be used for food preservation.
Reference: [ MOREIRA S P, CARVALHO W M, ALEXANDRINO A C, et al. Freshness retention of minimally processed melon using different packages and multilayered edible coating containing microencapsulated essential oil[J].International Journal of Food Science and Technology, 2014, 49 (10) :2192-2203]
A supplementary description "It can be used for food preservation, especially for food with high fat content, and is expected to be widely used in food field instead of chemically synthesized antioxidants." has been added to the corresponding position in the article. Line 768

Reviewer 3 Report
The manuscript "molecules-1669537" describes physicochemical and cellular studies of hyperoside with three common cyclodextrins. The paper needs a decent rewrite and a thorough language correction. Cell-based and antioxidant assays are not in my area of expertise, so I will not be rating this section. However, I would like to point out that Figures 11 and 12 are completely unreadable.
Back to the physicochemical tests. The authors almost always omit comparisons of complexes with physical mixtures, moreover the corresponding systems for the other two cyclodextrins are omitted both in the results and in the discussion. The paper contains a number of imprecise and jargon terms, and there is a lack of explanation of abbreviations in many places. Quite often there is no logical flow of the statement, often there are references to the literature but in my opinion not on topic, too general.
Regarding the abstract: all cyclodextrins studied should be listed. The word cyclodextrins has a very broad meaning and it is not necessary to add the term "and its derivatives". L. 14: ...cyclodextrins and its derivatives...". A grammatical error is also evident here: instead of "their", it is "its".
Introduction: It is mandatory to present the structural formula of hyperoside. L. 29: Since the authors add the word "complex" they mean that it is not a chemical individual but some mixture? The terms "oral permeability" and "oral bioavailability" are unfortunate. They need to be corrected. If the glucopyranose fragment is part of a larger molecule, then indicating a torsion mark for the cyclodextrin fragment is meaningless (l. 42: (+)-glucopyranose...). The abbreviation "Glc" was introduced without prior explanation (l. 43 and 44). The sentence in l. 44-48 is incorrectly worded. L. 55 and 56: "hydroxyl" and "amino" here used unprofessionally, jargon-wise. L. 60 and 61: "... the intramolecular hydroxyl group of cyclodextrin is closed". This is wrong, it is too short to explain the phenomenon. The improvement in solubility after cyclodextrin ether formation is due to the prevention of intramolecular hydrogen bond formation. These bonds are present in native cyclodextrin and thus hydration of these molecules is less readily.
The abbreviation "ICs" is not explained.
The sentences (l. 68-75) are unnecessary. I do not see a logical sequence with the previous section of the introduction. In l. 77 no literature reference is given. It is very important to pose the scientific problem that the authors want to solve in their paper. Unfortunately there is a lack of any sign of the authors' main idea.
In l. 139: "... ultrasonic inclusion..." another jargon term, imprecise.
FTIR: At what resolution was the scan performed? The authors go on to show results to two decimal places. IR spectra are not recorded with this accuracy, so the numbers need to be corrected in the discussion.
In l. 279 there should be a citation to the computer program. Links are also added to computer packages.
Results (l. 281): No introduction to the discussion at all. Which method? The reader is confused about what will be discussed.
L. 337: crystalline molecules and l. 338: amorphous molecule. The molecules themselves have no crystallographic form. These terms should be corrected.
Fig. 3: Curves for pure cyclodextrins and physical mixtures are not shown.
L. 346: "a chromatographic technology". This is not a technology but a method.
L. 372 and 373: "XRD test results of each monomer, ICs and physical mixtures were shown in Figure 5." This is not true. The results for pure cyclodextrins and mixtures are not shown. The sentence in l. 381-382 is unnecessary, it makes no sense here.
Figure 6: And for the other complexes why were the results not shown?
Similarly, in the NMR section, results and discussion for the other two complexes were omitted. Why? Please make a table, where the chemical shifts will be compared. Figure 7 as unreadable and not contributing relevant data should be placed in supplementary material.
The sentence in l. 454-456 is very obvious. It should be removed.
Figure 8: Results for mixtures are not shown.
L. 479-482: Citations 63 and 64 are unnecessary, there is no link to these studies.
Figure 9b and 9c this is unreadable.
Figure 10: Three points forming a straight line - not very convincing. These are not enough experimental points.
The conclusions are very poor. No summary of any kind, no comparison of results. There is basically nothing in these conclusions.
I have raised many serious objections in this review. If the authors do not cover the missing results, discussion, I will be in favor of rejecting the paper.
Author Response
Dear reviewer,
Thank you very much to review my manuscript (molecules-1669537), and give me some suggestions. These comments are valuable and very helpful. We have read through comments carefully and have made corrections. Based on the instructions you provided, we uploaded the field of the revised manuscript. Revisions in the text are shown using red highlight for additions, and strike through font for deletions. The responses to your comments are marked in red and presented following in this word.
We would love to thank you for allowing us to resubmit a revised copy of the manuscript and we highly appreciate your time and consideration.
Yours sincerely,
Xinyu Zhang
General Comments:
Comment 1: The paper needs a decent rewrite and a thorough language correction.
Response 1: We have rewritten the manuscript for inappropriate expressions and completed language correction.
Comment 2: Figures 11 and 12 are completely unreadable.
Response 2: We adjusted the sharpness of the picture and enlarged it. Figures 11 and 12
Comment 3: Back to the physicochemical tests. The authors almost always omit comparisons of complexes with physical mixtures, moreover the corresponding systems for the other two cyclodextrins are omitted both in the results and in the discussion.
Response 3: We supplement the TGA, DSC and XRD results of three cyclodextrins and physical mixtures of three systems. Because we originally wanted to compare Hyperoside with various inclusion complexes, other test results were not included in the original manuscript. However, we think your suggestion is very reasonable, so we have added the experimental results in the article.
Comment 4: Regarding the abstract: all cyclodextrins studied should be listed. The word cyclodextrins has a very broad meaning and it is not necessary to add the term "and its derivatives". L. 14: ...cyclodextrins and its derivatives...". A grammatical error is also evident here: instead of "their", it is "its".
Response 4: Sentence "the inclusion complexes of Hyp with cyclodextrins and its derivative were prepared by ultrasonic method" is changed to Sentence three inclusion complexes (ICs) of Hyp with 2-hydroxypropyl-β-cyclodextrin (2HP-β-CD), β-cyclodextrin (β-CD), and Methyl-β-cyclodextrin (M-β-CD) were prepared using the ultrasonic method, respectively.” Line 14
Comment 5: Introduction: It is mandatory to present the structural formula of hyperoside.
Response 5: I think your suggestion is very necessary, which will be conducive to a clearer understanding of the research object. Therefore, I added the chemical structure diagram of Hyperoside in the introduction part, as shown in Figure 1.
Comment 6: Line29: Since the authors add the word "complex" they mean that it is not a chemical individual but some mixture?
Response 6: Referring to relevant literature, the word "Inclusion Complex" is used in this article. Hyperoside was inserted into the cavity of cyclodextrin to form a new substance. But the new substance is still made up of the above two kinds of material, so we think the use of the word " Inclusion Complex " is relatively reasonable.
Reference: [Tian Bingren et al. The application and prospects of cyclodextrin inclusion complexes and polymers in the food industry: a review[J]. Polymer International, 2020, 69(7) : 597-603.]
[Romina L. Abarca et al. Characterization of beta-cyclodextrin inclusion complexes containing an essential oil component[J]. Food Chemistry, 2016, 196 : 968-975.]
Comment 7: The terms "oral permeability" and "oral bioavailability" are unfortunate. They need to be corrected.
Response 7: The terms "oral permeability" and "oral bioavailability" were changed to the terms "permeability" and " bioavailability". Line 41and 43
Comment 8: The abbreviation "Glc" was introduced without prior explanation (l. 43 and 44).
Response 8: The word "Glc" was replaced with the word "glucose". Line 54
Comment 9: The sentence in l. 44-48 is incorrectly worded.
Response 9: Sentence " The molecular structure of cyclodextrin is a truncated cone oligosaccharide with a hydrophobic cavity[20], which can form water-soluble host-guest inclusion complexes or assemble complex supramolecular systems through various non-covalent interactions with many organic or inorganic molecules, such as van der Waals force, hydrogen bonding and hydrophobicity[21]." is changed to Sentence “The molecular structure of cyclodextrin is truncated conical oligosaccharides with hydrophobic cavities [20]. It can form water-soluble host-guest ICs or assemble complex supramolecular systems with many organic or inorganic molecules through various non-covalent interactions, such as van der Waals forces, hydrogen bonding, and hydrophobicity [21].” Line 55
Comment 10: Line 55 and 56: "hydroxyl" and "amino" here used unprofessionally, jargon-wise.
Response 10: Sentence " Therefore, it is better to use β-cyclodextrin (β-CD) with hydroxyl or cyclodextrin derivatives with amino in the molecule for encapsulation " is changed to Sentence “Therefore, it is better to select β-CD with hydroxyl group or cyclodextrin with amino groups for encapsulation” Line 70
Comment 11: If the glucopyranose fragment is part of a larger molecule, then indicating a torsion mark for the cyclodextrin fragment is meaningless (l. 42: (+)-glucopyranose...).
Response 11: Cyclodextrin is a series of cyclic oligosaccharides produced by amylose under the action of cyclodextrin glucosyltransferase produced by Bacillus, usually containing 6-12 D-glucopyranose units. Among them, molecules containing 6, 7 and 8 glucose units, known as alpha-, beta- and gama-cyclodextrin, are more studied and of great practical significance. According to X-ray crystal diffraction, infrared spectroscopy and nuclear magnetic resonance spectroscopy, it is determined that each D (+)-glucopyranose of cyclodextrin molecule is chair conformation. Each glucose unit is ringed by 1, 4-glycosidic bond. Cyclodextrins are not cylindrical molecules but slightly cone-shaped rings because the glycosidic bonds that connect the glucose units do not rotate freely.
Comment 12: L. 60 and 61: "... the intramolecular hydroxyl group of cyclodextrin is closed". This is wrong, it is too short to explain the phenomenon. The improvement in solubility after cyclodextrin ether formation is due to the prevention of intramolecular hydrogen bond formation. These bonds are present in native cyclodextrin and thus hydration of these molecules is less readily.
Response 12: Sentence “After methylation of cyclodextrin, the intramolecular hydroxyl group of cyclodextrin is closed, which improves the stability of inclusion with drugs, and has a good solubilization effect on insoluble drugs” is changed to Sentence “Relevant literature shows that the methylated cyclodextrin can improve the stability of ICs and has a good solubilization effect on insoluble drugs [32]”. Line 75
Comment 13: The abbreviation "ICs" is not explained.
Response 13: A supplementary description "It can form water-soluble host-guest ICs or assemble complex supramolecular systems with many organic or inorganic molecules through various non-covalent interactions " has been added to the corresponding position in the article. Line 57
Comment 14: The sentences (l. 68-75) are unnecessary. I do not see a logical sequence with the previous section of the introduction.
Response 14: Sentences (L.68-75) illustrate the previous statement that the pharmacological activity of a drug is increased by encapsulation.
Comment 15: In l. 77 no literature reference is given. It is very important to pose the scientific problem that the authors want to solve in their paper. Unfortunately, there is a lack of any sign of the authors' main idea.
Response 15: Sentence “At present, there are few studies on inclusion of Hyp and cyclodextrin, so it is of great significance to systematically evaluate its inclusion complexes to expand its commercial application in the future.” is changed to Sentence “To date, there is no literature on the ICs of Hyp and cyclodextrin, so it is of great significance to systematically evaluate these ICs in this study so as to expand their applications in medicine, food and other fields in the future.”. Line 94
Comment 16: In l. 139: "... ultrasonic inclusion..." another jargon term, imprecise.
Response 16: Sentence “The absorbance value was measured at 360 nm after ultrasonic inclusion at 25°C for 60 min.” is changed to Sentence “The absorbance value was measured at 360 nm after ultrasonic mixing at 25 °C for 60 min with △ABS as the ordinate. △ABS is the distinction in the absorbance of Hyp with CDs or not, and the ratio of [Hyp]/([Hyp]+[CD]) was taken as the abscissa.”. Line 592
Comment 17: FTIR: At what resolution was the scan performed? The authors go on to show results to two decimal places. IR spectra are not recorded with this accuracy, so the numbers need to be corrected in the discussion.
Response 17: Both the data in the text and the data in the figure are presented as integers. Section 2.2.5; Figure 7
Comment 18: L. 337: crystalline molecules and l. 338: amorphous molecule. The molecules themselves have no crystallographic form. These terms should be corrected.
Response 18: Sentence “Different from crystalline molecules, the dissolvement of an amorphous molecule does not need any energy to decompose lattice” is changed to Sentence “In contrast to materials with crystal structure, the dissolving of materials with amorphous structure does not need any energy to decompose the lattice”. Line 191
Comment 19: Fig. 3: Curves for pure cyclodextrins and physical mixtures are not shown.
- 372 and 373: "XRD test results of each monomer, ICs and physical mixtures were shown in Figure 5." This is not true. The results for pure cyclodextrins and mixtures are not shown.
Response 19: The results of TGA, DSC, and XRD of cyclodextrins and three physical mixtures are added to the text. Section 2.2.2 and 2.2.4; Figure 4 and 6
Comment 20: L. 346: "a chromatographic technology". This is not a technology but a method.
Response 20: Sentence “Thin layer chromatography is a chromatographic technology applied to segregate mixtures.” is changed to Sentence “Thin-layer chromatography is a chromatographic method applied to segregate mixtures.”. Line 205
Comment 21: The sentence in l. 381-382 is unnecessary, it makes no sense here.
Response 21: Sentence “Zheng[60] obtained a similar conclusion in the study of ICs formed by α-lipoic acid and 2H-β-CD.” has been deleted. Line 246
Comment 22: Figure 6: And for the other complexes why were the results not shown?
Response 22: We initially performed scanning electron microscopy, thermogravimetry, thermal analysis, thin layer chromatography and X-ray diffraction analysis on these three inclusion complexes. The results showed that the three cyclodextrins formed inclusion complexes with Hyperoside. Subsequently, we carried out Solubility test, among which the solubilization effect of Hyp-2H-β-CD was selected as the research object of the follow-up test. Therefore, we only performed complementary experiments on it in the FTIR and 1H NMR analyses.
Comment 23: Similarly, in the NMR section, results and discussion for the other two complexes were omitted. Why? Please make a table, where the chemical shifts will be compared.
Figure 7 as unreadable and not contributing relevant data should be placed in supplementary material.
Response 23: In order to present the test results more clearly, we presented the test results in the form of tables, as shown in Table 1 and 2. The spectrogram was moved into the supplementary material, as shown in Figure S1.
A supplementary description “After the IC was formed, the chemical shift of H of the Hyp changed to varying degrees, as shown in Table 1 and 2. This means that the Hyp molecule successfully included into the 2H-β-CD cavity. The chemical shift of H of the 2H-β-CD also changed, which may be caused by the formation of hydrogen bond between the molecule of 2H-β-CD and Hyp. In addition, the chemical shift value of H-5 in 2H-β-CD cavity is larger than that of H-3. In the stereoscopic structure of cyclodextrin, H-3 and H-5 are located in the cavity of cyclodextrin, and H-3 is located at the large mouth end and H-5 is located at the small mouth end. It is suggested that the Hyp enters the cavity of 2H-β-CD molecule from the small mouth end.” have been added to the corresponding position in the article. Line 298
Comment 24: The sentence in l. 454-456 is very obvious. It should be removed.
Response 24: Sentence “Its internal hydrophobic as well as external hydrophilic features could make insoluble drug molecules enter its inner cavity and form IC[62], thus greatly improving the solubility of Hyp in water.” has been deleted. Line 339
Comment 25: Figure 8: Results for mixtures are not shown.
Response 25: The purpose of this study is to prepare an inclusion complex that can achieve good solubilization of Hyp. The research focuses on inclusion complexes rather than physical mixture. In addition, a series of characterization results showed that the Hyp in the physical mixture did not enter the molecules of cyclodextrin, so we think it is not necessary to detect the water-solubility of the physical mixture. We sincerely hope that you can understand our ideas.
Comment 26: Figure 9b and 9c this is unreadable.
Response 26: Figure 9 has been redone.
Comment 27: Figure 10: Three points forming a straight line - not very convincing. These are not enough experimental points.
Response 27: The test scheme was designed with reference to literature [69] and adjusted appropriately according to the actual situation.
Reference: [69] Nguyen, T.A.; Liu, B.G.; Zhao, J.; Thomas, D.S.; Hook, J.M. An investigation into the supramolecular structure, solubility, stability and antioxidant activity of rutin/cyclodextrin inclusion complex. FOOD CHEMISTRY 2013, 136, 186-192, doi:10.1016/j.foodchem.2012.07.104.
Comment 28: The conclusions are very poor. No summary of any kind, no comparison of results. There is basically nothing in these conclusions.
Response 28: The conclusion part has been rewritten.
- Conclusions
Hyperoside exhibits a series of pharmacological activity, but its application is limited because of its poor water solubility and low bioavailability. In this paper, the ICs of Hyp and cyclodextrin were successfully prepared with 1:1 stoichiometric. Evidence shows that the ICs with 2H-β-CD shows the higher water solubility compared with Hyp-β-CD or Hyp-M-β-CD. Thermodynamic studies demonstrated that the thermal stability of the three ICs is obviously stronger than that of Hyp, and the inclusion of Hyp with 2H-β-CD was a spontaneous exothermic process. According to SEM images and 1H NMR analysis, the Hyp molecules are inserted into the 2H-β-CD cavity to form the complex structure. The structure can protect Hyp from heat and prevent degraded for high temperature, and significantly enhanced its antioxidant capacity. The formation of this structure may be related to the presence of hydroxypropyl group in 2H-β-CD can significantly enhance its capacity to entrap Hyp. This inclusion technology could be especially helpful for increasing the application of Hyp in functional foods, beverages, and drugs. It can be used for food preservation, especially for food with high fat content, and is expected to be widely used in food field instead of chemically synthesized antioxidants.

Round 2
Reviewer 3 Report
The manuscript has been revised accordingly, and I recommend its publication.
This manuscript is a resubmission of an earlier submission. The following is a list of the peer review reports and author responses from that submission.